# Effect of Growth Stages on Anthocyanins and Polyphenols in the Root System of Sweet Potato

**DOI:** 10.3390/plants12091907

**Published:** 2023-05-08

**Authors:** Yuno Setoguchi, Soya Nakagawa, Ryo Ohmura, Saki Toshima, Hyungjun Park, Yosuke Narasako, Tomonari Hirano, Motoyasu Otani, Hisato Kunitake

**Affiliations:** 1Graduate School of Agriculture, University of Miyazaki, Miyazaki 889-2192, Japan; 2Division of Symbiotic System, National Institute for Basic Biology, Okazaki 444-8585, Japan; 3Interdisciplinary Graduate School of Agriculture and Engineering, University of Miyazaki, Miyazaki 889-2192, Japan; 4Kushima Aoi Farm Co., Ltd., Miyazaki 889-3531, Japan; 5Faculty of Agriculture, University of Miyazaki, Miyazaki 889-2192, Japan; 6Research Institute of Bioresources and Biotechnology, Ishikawa Prefectural University, Nonoichi 921-8836, Japan

**Keywords:** acylated anthocyanin, caffeoylquinic acid, HPLC, pigment, histological observation

## Abstract

The storage roots of purple-fleshed sweet potato contain a variety of anthocyanins and polyphenols. Little is known about changes in the total content and composition of anthocyanins and polyphenols in the early growth stages of the root system. In this study, we investigated the changes in anthocyanins and polyphenols in the root system of purple-fleshed sweet potato cultivars at 15, 30, 45, and 60 days after transplant (DAT). Unexpectedly, the highest percentage of acylated anthocyanins in three purple-fleshed cultivars among all growth stages was at 15 DAT. On the other hand, the total polyphenol content in the early growth stages of the root system increased rapidly toward 45 DAT, just before the beginning of storage root enlargement, and then decreased rapidly as the storage roots began to enlarge. These data indicate that the early growth stage of the root system is a critical time. This timing may present a strategy to maximize the accumulation of polyphenols with high antioxidant activity, as well as acylated anthocyanins, to protect against abiotic and biotic stresses.

## 1. Introduction

*Ipomoea batatas* (L.) Lam, more commonly sweet potato, is a dicotyledonous plant belonging to the family Convolvulaceae [1]. In terms of life cycle, sweet potato is classified as perennial, but is cultivated as an annual crop throughout the tropics and warm, temperate regions of the world. Its natural storage roots have high contents of starch and fibers, as well as minerals, vitamins, and polyphenols [2]. Particularly, purple-fleshed sweet potato cultivars contain bioactive compounds such as polyphenols and anthocyanins, making them attractive to consumers who want to eat healthy foods.

Thousands of polyphenols have been identified in higher plants, and several hundred are found in edible plants [3]. Polyphenols are secondary metabolites in plants and are generally involved in defense against UV light or pathogen attacks [4]. In sweet potato, the main polyphenols are caffeoylquinic acids. The raw storage roots contain caffeic acid, chlorogenic acid (5-caffeoylquinic acid), a single molecule of caffeic acid bound to quinic acid, and dicaffeoylquinic acid (3,4-dicaffeoylquinic acid, 3,5-dicaffeoylquinic acid, 4,5-dicaffeoylquinic acid), which is a double molecule of quinic acid [5].

The Andean region of South America is the origin of many crops, and purple-fleshed sweet potato has also been reported to originate from this region [6]. The storage roots of purple-fleshed sweet potato contain large amounts of anthocyanins. Sweet potato anthocyanins exist in non-, mono-, and di-acylated forms of cyanidin and peonidin, and have been specifically acylated with ferulic acid, caffeic acid, and hydroxybenzoic acid moieties [7]. In addition to the basic research on anthocyanin biosynthesis, studies on the acculturation patterns of polyphenols and anthocyanins in storage roots have revealed important information for cultivation and breeding [8,9]. There have been several reports of changes in polyphenol and anthocyanin contents in developing tubers of purple- and orange-fleshed sweet potato [10,11,12].

Nakagawa et al. (2021) reported that the percentages of anthocyanins that were acylated in purple-fleshed cultivars were already high in the early stage of tuber growth (60 days after transplant [DAT]) and that the changes in total anthocyanin content differed among the three characterized cultivars. Surprisingly, the percentages of acylated anthocyanins, as well as the peonidin/cyanidin ratios of the three purple-fleshed cultivars, were determined mostly in the early stage of tuber growth, although there were differences among cultivars [12]. However, the content and composition of anthocyanins and polyphenols of the root system before tuber growth have not been reported so far.

Therefore, we investigated the changes in polyphenol and anthocyanin content and composition in the root systems of four sweet potato cultivars in the early growth stages. In addition, we discussed why acylated anthocyanins and polyphenols need to accumulate in the root systems of purple-fleshed sweet potato in the early growth stage.

## 2. Results

### 2.1. Histological Observation of Root System

Roots were already visible in all cultivars at 15 DAT. In the early growth stages, the root system consisted of adventitious and lateral roots. Most adventitious roots originated from the adventitious root primordium in the leaf gap at the nodes of the nurseries, while lateral roots were differentiated from any part of the adventitious root (Figure 1A). The adventitious roots then mainly enlarged and became storage roots (Figure 1B), but differences in the timing of enlargement were observed among the cultivars. Purple accumulation was already observed at the base of the adventitious roots of ‘Akemurasaki’ from 15 DAT (Figure 1C), and those roots were entirely dark purple at 30 DAT (Figure 1D). On the other hand, no pigment accumulation was observed in most of the lateral roots.

Histological observation in the root system confirmed that general tissue differentiation of sweet potato was confirmed (Figure 2). As shown in the figure, primary cambium, protoxylem, meristem cells around the vessel, and periderm can be seen. Sweet potato is a xylem hypertrophic growth type crop. At 15 DAT, the primary cambium of the adventitious roots clearly surrounded the vessels, and the vessels were randomly differentiated within it (Figure 2A). In particular, the differentiation of meristematic cells around the vessel at 45 DAT was observed in ‘Purple Sweet Lord’, where storage roots were enlarged from a relatively early stage (Figure 2B). In addition, as storage roots grew larger, a periderm formed, covering the outer layer of the storage roots (Figure 2C,D).

### 2.2. Change in Anthocyanin Content and Composition

The changes in anthocyanin composition during the growth of the root system are shown in Table 1, Table 2, Table 3 and Table 4, along with cross sections of the roots of the four cultivars. Moreover, changes in the anthocyanin content during root system growth in the three purple-fleshed cultivars and the yellow-fleshed control cultivar, ‘Kokei No. 14’, were investigated, and the results are summarized as percentages of acylated anthocyanin and peonidin/cyanidin ratios during root growth (Figure 3). The total anthocyanin content and composition during the early growth of the root system varied among the cultivars (Figure 3, Table 1, Table 2, Table 3 and Table 4).

A small amount of anthocyanin was accumulated from 15 DAT in the ‘Kokei No. 14’, a yellow-fleshed cultivar with purple skin, tested as a control. This cultivar showed pigmentation only in the peel at all times investigated, as seen in cross sections of the root (Table 1). Its anthocyanin composition was dominated by acylated anthocyanins such as YGM-6, but only peonidin 3-hydroxybenzoic acid 5-glucoside showed the highest content at harvest time. The content was 1.3 mg·100 g^−1^ FW, which accounted for about 68% of the total anthocyanin content at harvest time (Table 1).

‘Akemurasaki’, a purple-fleshed cultivar of the peonidin type, accumulated several anthocyanins at 15 DAT, and the observation of root cross sections showed that pigmentation started externally from the peel and cortex layers of the adventitious root (Table 2). Total anthocyanin content at harvest was significantly higher, at 501.2 mg·100 g^−1^ FW, than at the other stages of root growth. The main anthocyanins in the root system at 45 DAT were YGM-5a (54.1 mg·100 g^−1^ FW), YGM-1a (46.5 mg·100 g^−1^ FW), and YGM-2 (20.3 mg·100 g^−1^ FW), but those at harvest time were YGM-5b (110.1 mg·100 g^−1^ FW), YGM-5a (73.2 mg·100 g^−1^ FW), and YGM-2 (71.7 mg·100 g^−1^ FW).

On the other hand, the same peonidin type, ‘Purple Sweet Lord’, showed the highest total anthocyanin content at 60 DAT (84.4 mg·100 g^−1^ FW), but the content decreased to 45.1 mg·100 g^−1^ FW at harvest time as the storage roots grew (Table 3). The main anthocyanins in the root system at 45 DAT were YGM-5a (6.8 mg·100 g^−1^ FW), YGM-1a (5.7 mg·100 g^−1^ FW), and YGM-2 (3.6 mg·100 g^−1^ FW), but those at harvest time were YGM-5b (10.5 mg·100 g^−1^ FW), YGM-6 (9.6 mg·100 g^−1^ FW), and YGM-5a (5.0 mg·100 g^−1^ FW). In the observation of root cross sections, pigment deposition was observed throughout the roots at 30 DAT.

‘Tanegashimamurasaki’, a purple-fleshed cultivar of the cyanidin type, accumulated few anthocyanins until 60 DAT and showed significantly higher values at harvest time (23.3 mg·100 g^−1^ FW). Pigment deposition was also not observed in root cross sections until 60 DAT, and was confirmed only at harvest time (Table 4). Peonidin 3-hydroxybenzoic acid 5-glucoside, which was characteristically observed in ‘Kokei No. 14’, was not accumulated at all.

Changes in the percentages of acylated anthocyanin and peonidin/cyanidin ratios during the growth of the root system were investigated (Figure 3). The percentages of acylated anthocyanins in all cultivars were highest at 15 DAT among all growth stages: 98.5% for ‘Akemurasaki’, 100% for ‘Purple Sweet Lord’, and 100% for ‘Tanegashimamurasaki’. As growth continued, the percentages decreased mildly but remained high (Figure 3A). On the other hand, the peonidin/cyanidin ratios in all cultivars were significantly higher at harvest time among all growth stages (Figure 3B).

### 2.3. Changes in Polyphenol Content and Composition

The total polyphenol content during the growth of the root system in the three purple-fleshed cultivars and ‘Kokei No. 14’ were investigated by HPLC analysis (Figure 4), and the percentage of each polyphenol among all polyphenols was evaluated (Figure 5). The changes in total polyphenol content in the root system showed similar trends for all cultivars, although there were some differences for ‘Purple Sweet Lord’. The total polyphenol content of the root system at 15 DAT was already almost the same as that of the storage roots at harvest, and then it increased rapidly at 45 DAT. For the four cultivars, the values at 45 DAT were approximately 1.4–4.9 times higher than the values at 15 DAT. In particular, a high polyphenol content of 307.3 mg·100 g^−1^ FW was recognized in the root system of ‘Akemurasaki’. Subsequently, the total polyphenol content of all cultivars decreased with root growth at harvest time. The decrease rate ranged from 28 to 85%.

In addition, five polyphenols, namely, 3,5-dicaffeoylquinic acid, chlorogenic acid, 3,4-dicaffeoylquinic acid, 4,5-dicaffeoylquinic acid, and caffeic acid, were detected in the four types of sweet potato cultivars, and 3,5-dicaffeoylquinic acid was the most common polyphenol at 15 DAT in the examined samples (66.3–73.2%: 3,5-dicaffeoylquinic acid content/total contents of five individual polyphenols). After that, the percentage of 3,5-dicaffeoylquinic acid decreased as the root system grew, and at harvest time, the percentage of chlorogenic acid was higher, accounting for 30 to 40% of the total.

## 3. Discussion

Sweet potato storage roots are highly nutritious and offer sensory diversity in taste, texture, and flesh colors, such as white-yellow, orange, and purple. In particular, the purple-fleshed sweet potato cultivars with attractive colors and high anthocyanin contents are considered special types in Asia and the Pacific islands. Furthermore, these polyphenols and anthocyanins in the storage roots of purple-fleshed sweet potato have diverse biological activities, including antioxidant, anti-inflammatory, anticancer, antidiabetic, and hepatoprotective effects [13].

On the other hand, polyphenols and anthocyanins in the root system of sweet potato are also known to be major components of resistance to biotic and abiotic stresses [14,15,16]. Yoshinaga et al. (2000) and Nakagawa et al. (2021) reported that the total polyphenol contents, the peonidin/cyanidin ratios, and the percentages of acylated anthocyanins in purple-fleshed cultivars were almost fixed in the early stage of tuber growth (around 60 DAT) and did not vary greatly after tuber growth began [10,12]. However, little information is available on the expression of polyphenols and anthocyanins in the root system prior to the growth of storage roots. In this study, we investigated the changes in polyphenol and anthocyanin content and composition in the root system of four sweet potato cultivars in the early growth stages.

The expression of anthocyanins in the early stage of adventitious root formation of ‘Akemurasaki’, which accumulates anthocyanins abundantly in storage roots, was investigated. The accumulation of anthocyanin was observed at the base of adventitious roots at 15 DAT. At 30 DAT, the pigmentation began in the outer cortex layer of the adventitious roots. Yoshinaga et al. (2000) reported that anthocyanin was detected in the young, thick roots of most purple-fleshed sweet potato clones at 21 DAT [10]. In the present study, anthocyanin accumulation occurred at an earlier stage of root system formation (15 DAT), and its expression patterns differed among cultivars. In particular, unlike other cultivars, which accumulate anthocyanins in the root system from an earlier stage, ‘Tanegashimamurasaki’, a purple-fleshed cultivar of the cyanidin type, accumulated few anthocyanins until 60 DAT and showed significantly higher values at harvest time than at the other time points. Recently, Ning et al. (2021) provided new insight into the regulatory network of anthocyanin biosynthesis in purple-fleshed sweet potato roots, demonstrating the interaction of different transcription factors in the regulation of anthocyanin biosynthesis [17]. The differences in the expression patterns of anthocyanins in sweet potato may involve the type of anthocyanin and the transcription factors that regulate them.

Anthocyanins of purple-fleshed sweet potato consist of mono- or di-acylated forms of cyanidin (YGM-1a, -1b, -2, and -3) and peonidin (YGM-4b, -5a, -5b, and -6) [18]. The present study showed, for the first time, that its anthocyanin composition varied with the growth stage of the root system. In other words, the anthocyanin composition of the two peonidin-type cultivars differed at harvest, but the major anthocyanins (YGM-5a, YGM-1a, and YGM-2) matched in the early growth of the root system at 45 DAT. In addition, only YGM-2 was accumulated in the cyanidin-type ‘Tanegashimamurasaki’ in the early growth of the root system.

The percentage of acylated anthocyanins was found to be high in the early growth stages of the root system. Acylation reactions of anthocyanins are generally reported to be the last step in the anthocyanin process [19]. Considering the biosynthetic pathway, it is expected that basic anthocyanins accumulate first, followed by a gradual increase in the proportion of acylated anthocyanins. However, contrary to our expectations, we found that the percentage of acylated anthocyanins was higher in the early growth stage of the root system than at the harvest stage. This result may suggest the need to promote acylation reactions and maintain higher anthocyanin content from the early growth stage of root formation than in mature storage roots. The root system of sweet potato is under constant stress and may protect itself from biotic and abiotic stress with acylated anthocyanins that accumulate from the early stage of root formation, as mentioned above.

Finally, the total polyphenol content of the root system was examined. Surprisingly, it increased rapidly toward 45 DAT and then decreased rapidly as storage roots began to enlarge. Previous studies have extensively investigated the anatomy of the sweet potato root system. Noh et al. (2010) described that the stele cells in the early root system are highly lignified and have weak vascular cambium activity [20]. Recently, He et al. (2021) reported that storage root formation in sweet potato is intricately regulated by a transcriptional regulatory network and suggested that the transition point from pre-swelling to storage roots is around the S10 stage, when the core regulators of the transcriptional regulatory network are activated [21]. Furthermore, Meng et al. (2022) reported that ammonium nitrogen treatment promoted *IbEXP* and suppressed the expression of the *KNOX1* gene, *Ibkn1*, and *Ibkn2*, which positively regulate cytokinin biosynthesis in the early stages of storage root formation (14–28 DAT); followed by increased gibberellic acid and decreased zeatin riboside content and phenylalanine ammonia lyase and peroxidase activities; and promote lignin synthesis in potential storage roots [22]. In this study, we consider the transition point from pre-swelling to storage roots to be 30 DAT. Therefore, it is assumed that flavonoid synthesis predominated over lignin synthesis at 45 DAT after the tipping point, resulting in the high polyphenol content at 45 DAT. It is also possible that the polyphenol content per 100 g FW decreased due to subsequent enlargement.

Moreover, plants utilize an antioxidant defense system to protect themselves from harmful reactive oxygen species, and plant stress tolerance correlates with the ability to scavenge and detoxify reactive oxygen species [23]. Reactive oxygen species are also thought to be involved in plant development and stress responses [24,25,26,27]. Superoxide dismutase (SOD) constitutes the first line of defense against ROS within plant cells [21]. Those authors found that SOD activity was higher in fibrous roots than in storage roots. This difference in SOD activity may be related not only to the total polyphenol content but also to the quenching of individual polyphenol contents such as chlorogenic acid and 3,5-dicaffeoylquinic acid obtained in this study.

As described above, this period before the beginning of storage root enlargement (around 30 DAT) is a critical time, and this timing may be a strategy to maximize the accumulation of polyphenols with high antioxidant activity, as well as acylated anthocyanins, to protect against abiotic and biotic stresses.

## 4. Materials and Methods

### 4.1. Plant Materials

Three purple-fleshed sweet potato cultivars, ‘Akemurasaki’, ‘Purple Sweet Lord’, and ‘Tanegashimamurasaki’, and the yellow-fleshed cultivar ‘Kokei No. 14’ as a control, were used in this experiment. ‘Akemurasaki’ and ‘Purple Sweet Lord’ have peonidin-type anthocyanins, while ‘Tanegashimamurasaki’ has the cyanidin type [12].

Stem cuttings of each cultivar were transplanted into culture containers (64 × 23 × 18.5 cm), filled with commercially available culture soil (NAFCO Co., Ltd., Fukuoka, Japan), and grown at the experimental farm at the Faculty of Agriculture, University of Miyazaki. To investigate the changes in anthocyanins and polyphenols during root growth, the root system containing adventitious roots and lateral roots was harvested at four developmental stages (15, 30, 45, and 60 DAT). The root systems of three plants were sampled and carefully washed with a spray of tap water. Storage roots of the four cultivars at harvest time (120–140 DAT) were used as a control.

### 4.2. Histological Observation of Root System

For the histological observation of the coloration inside the roots, raw materials harvested at the four developmental stages were used. Roots at each stage were sectioned freehand using a razor blade. Sections were observed under an S2 × 7 stereomicroscope (Olympus, Tokyo, Japan).

Fixed root materials were used for the histological observation of the vascular bundle development inside the root. Roots of each stage were selected as enlarged and immediately fixed in Farmer’s solution (ethanol/acetic acid = 3:1) and degassed in a vacuum desiccator by reducing the pressure to a non-boiling level. Fixed roots were sectioned thinly using a razor blade, and these sections were double stained with safranin and fast green. Briefly, these sections were washed with water, stained with safranin (1% safranin in 50% ethanol) for 10 s, and then stained with fast green (0.2% fast green in 95% ethanol) for 30 s. Stained sections were washed with tap water and observed using the SZX7 stereomicroscope.

### 4.3. Determination of Anthocyanin Content and Composition

Samples from five growth stages (15, 30, 45, 60 DAT, and harvest time) were analyzed using HPLC to investigate changes in anthocyanin content and composition during the growth stages of the root system.

All samples of roots were frozen to a temperature of −30 °C and then freeze dried in an FDU-1100 lyophilizer and a DRC-1000 chamber (Tokyo Rikakikai, Tokyo, Japan). The freeze-dried samples were powderized with a B-400 pulverizer (Nihon Buchi, Tokyo, Japan) and stored at −30 °C until used for experiments.

Total anthocyanin content and composition were also determined by high-performance liquid chromatography (HPLC) to separate individual anthocyanins in each sample [28]. Freeze-dried root powder (0.2 g) was extracted with 3 mL of extraction solution (methanol: water/trifluoroacetic acid = 40:60:0.5) in centrifuge tubes, and the tubes were placed in ultrasonic (US CLEANER, As One, Osaka, Japan) for 5 min at 37 °C. The mixtures were then centrifuged at 1500× *g* for 10 min, and an aliquot of the upper layer was taken and filtered using a 0.20 μm syringe filter before being analyzed by HPLC to assess the anthocyanins.

The extracts were analyzed by reverse-phase HPLC using the Prominence LC solution system (Shimadzu, Kyoto, Japan) with a Kinetex column (Shimadzu). The chromatographic conditions were as follows: solvent A, 0.6% (*v*/*v*) formic acid; solvent B, 50% (*v*/*v*) acetonitrile and 0.6% (*v*/*v*) formic acid; column temperature, 30 °C; detection at 520 nm; and flow rate, 0.6 mL·min^−1^. The column was equilibrated with 20% B before use. The binary gradient was as follows: 20% B (0–50 min), 50% B (50–55 min), and 20% B (55–60 min). The retention times and spectra used to identify anthocyanins were compared with pure standards of peonidin 3-caffeoyl-feruloyl-sophoroside-5-glucoside (YGM-6). Other putative anthocyanin peaks were presumed following Nakagawa’s study (2021). To investigate the content of each individual anthocyanin, three concentrations of YGM-6 were prepared at 49.52, 12.38, and 1.238 μM; 10 μL was injected as an external standard. Each individual anthocyanin was quantified using the peak area recorded at 520 nm to construct the calibration curve for YGM-6. In addition, the total anthocyanin content was defined as the sum of the individual anthocyanin contents. Moreover, monoacylated anthocyanin content was calculated as the sum of the individual monoacylated anthocyanin contents, and diacylated anthocyanin content was calculated as the sum of the individual diacylated anthocyanin contents. The measurements were replicated three times.

### 4.4. Determination of Polyphenol Content and Composition

Samples from five growth stages were analyzed using HPLC to investigate changes in polyphenol content and composition during the growth stages of the root system.

The total polyphenol content and composition were measured using HPLC and were used to separate and determine individual polyphenols in the sample. Freeze-dried root powder (0.02 g) was dissolved in 5 mL of 80% (*v*/*v*) methanol and passed through a 0.20 μm membrane filter (Sartorius, Göttingen, Germany) for analysis.

The extracts were analyzed by reverse-phase HPLC using the Prominence LC solution system with an ODS-3 column (Shimadzu). The chromatographic conditions were as follows: solvent A, 100% ethanol; solvent B, 20 mM KH2PO4 (pH 2.4); column temperature, 40 °C; detection at 320 nm; and flow rate, 1.0 mL·min^−1^. The binary gradient was as follows: 85–68% B (0–12 min), 68% B (12–15 min), 50–55% B (15–20 min), and 85% B (20–29 min). Retention times and spectra were compared with pure standards of chlorogenic acid, caffeic acid, 3,4-dicaffeoylquinic, 3,5-dicaffeoylquinic acid, and 4,5-dicaffeoylquinic acid. The results are expressed as mg·100 g^−1^ FW. The measurements were replicated three times.

### 4.5. Statistical Analysis

All experimental results are represented as means ± standard deviation and experiments were performed in triplicate. The results were evaluated for statistical significance using Tukey’s multiple range test. Excel and Excel statistics were used for statistical processing.

## 5. Conclusions

We investigated the changes in polyphenol and anthocyanin content and composition in the root systems of four sweet potato cultivars, including ‘Kokei No. 14’ as a control, in the early growth stages. The changes in anthocyanin and polyphenol content and composition early in the root system formation (0–60 DAT) were greater than the changes in storage root growth (60 DAT—harvest), although there were differences among the cultivars. In particular, the changes in the total polyphenol contents of the root systems just prior to enlargement were remarkable. Considering the early growth stage of root systems may be a strategy to maximize the accumulation of polyphenols with high antioxidant activity, as well as acylated anthocyanins, to protect against abiotic and biotic stresses. To the best of our knowledge, this is the first report of this finding in sweet potato, and it is noteworthy from not only a nutritional point of view but also from an agronomic point of view.

In recent years, volatile organic compounds (VOCs) produced by microorganisms have shown promise as environmentally safe fumigants for the control of postharvest diseases. In addition to observing direct antifungal activity, Gong et al. (2022) found that VOCs can also trigger the defense response of sweet potato and that VOC treatment increased the contents of antioxidant enzymes and total flavonoids in sweet potato [29]. In the future, the polyphenol and acylated anthocyanin content and composition of sweet potato root systems, as well as the timing of their expression, may become important indicators of disease resistance.

## Figures and Tables

**Figure 1 plants-12-01907-f001:**
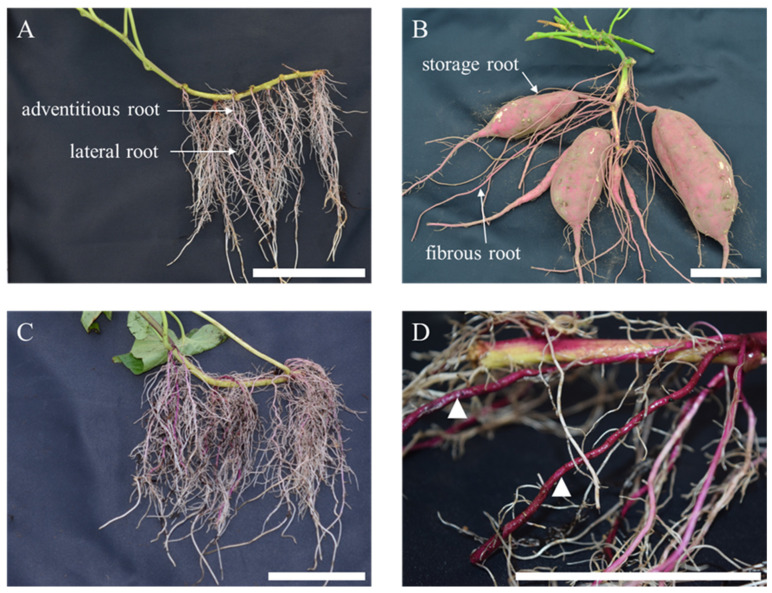
Root system of sweet potato. (**A**) ‘Kokei No. 14’ at 15 days after transplant (DAT), (**B**) ‘Kokei No. 14’ at harvest time, (**C**) ‘Akemurasaki’ at 15 DAT, (**D**) ‘Akemurasaki’ at 30 DAT. Arrow heads indicate adventitious roots with anthocyanin accumulation, bar = 10 cm.

**Figure 2 plants-12-01907-f002:**
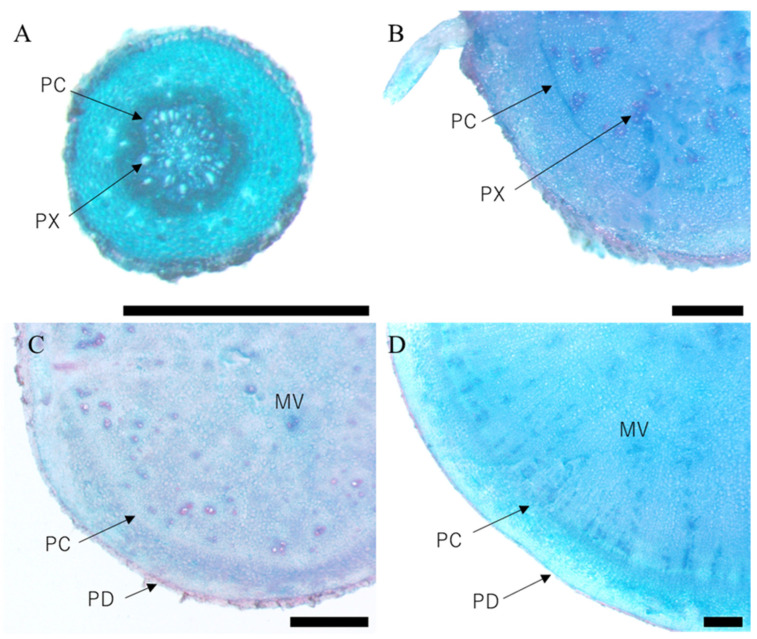
Histological observation of the vascular bundle in root system of ‘Purple Sweet Lard’. (**A**) 15 days after transplant DAT, (**B**) 30 DAT, (**C**) 45 DAT, (**D**) 60 DAT. PC: primary cambium, PX: protoxylem, MV: meristem cells around the vessel, PD: periderm, bar = 1 mm.

**Figure 3 plants-12-01907-f003:**
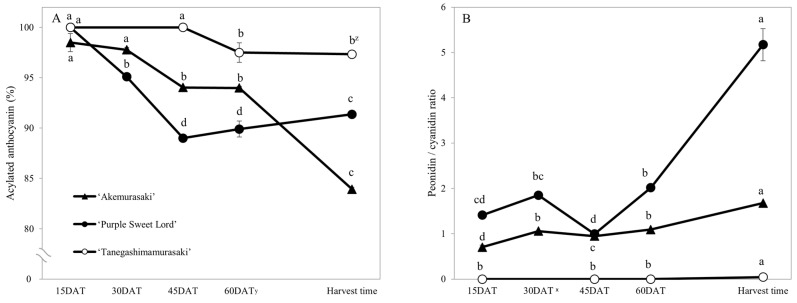
Effects of root growth stages on the percentage of acylated anthocyanin and peonidin/cyanidin ratio. (**A**): Change in the percentage of acylated anthocyanin during root growth of three purple-fleshed sweet potato variants. (**B**): Change in peonidin/cyanidin ratio during tuber development of three purple-fleshed sweet potato variants. Each datum represents mean ± S.D. (*n* = 3). ^z^ Different letters beside each set of 3 cultivars represent significant differences at 5% level as determined by Tukey’s multiple range test (*n* = 3). ^y^ DAT: Days after transplant; nurseries of four cultivars were transplanted to Miyazaki research field on 29 May 2021. ^x^ The peonidine/cyanidine ratio at 30 DAT in ‘Tanegashimamurasaki’ is not shown because anthocyanins were not detected.

**Figure 4 plants-12-01907-f004:**
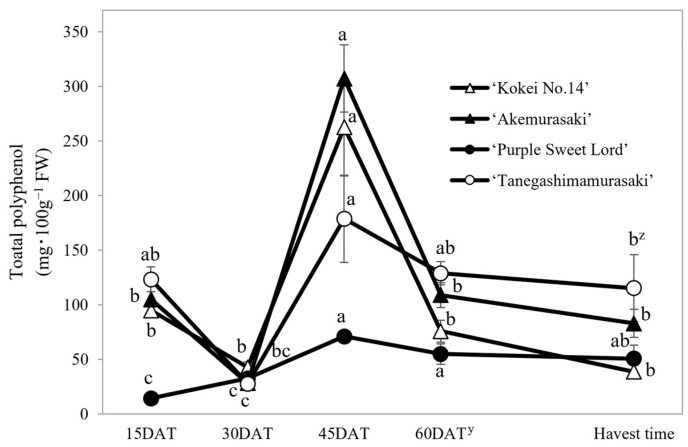
Effect of growth stages on the total polyphenol content in root system of four sweet potato cultivars. Each datum represents mean ± S.D. (*n* = 3). ^z^ Different letters beside each set of 4 cultivars represent significant differences at 5% level as determined by Tukey’s multiple range test (*n* = 3). ^y^ DAT: Days after transplant; nurseries of four cultivars were transplanted to Miyazaki research field on 29 May 2021.

**Figure 5 plants-12-01907-f005:**
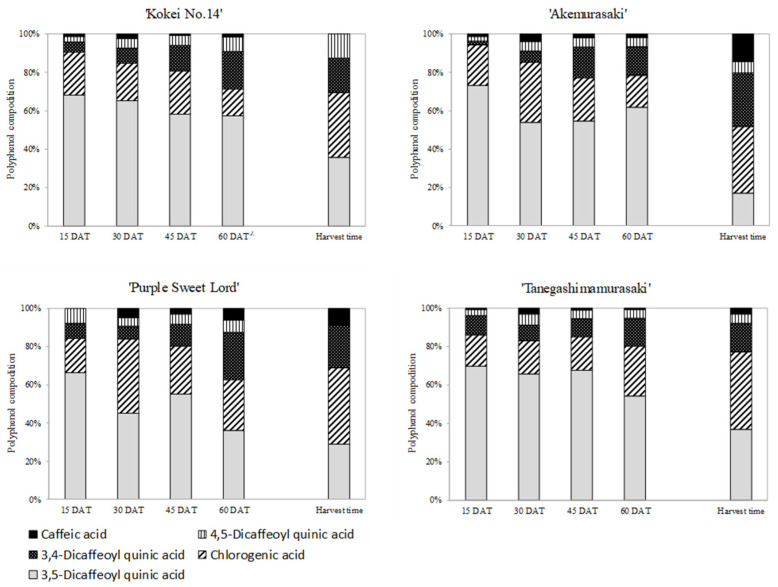
Effects of growth stages on the percentage of caffeic acid, 4,5-Dicaffeoyl quinic acid, 3,4-Dicaffeoyl quinic acid, chlorogenic acid, and 3,5-Dicaffeoyl quinic acid in root system of four sweet potato cultivars. ^z^ DAT: Days after transplant; nurseries of four cultivars were transplanted to Miyazaki research field on 29 May 2021.

**Table 1 plants-12-01907-t001:** Effect of growth stages on anthocyanin content and composition in root system of ‘Kokei No. 14’.

DAT ^y^	15	30	45	60	Harvest Time
	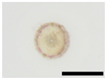	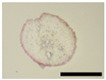	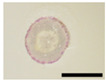	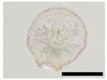	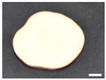
	Anthocyanin content (mg·100 g^−1^ FW)				
Cya 3-sop-5-glu	0.0 ± 0.0 NS ^x^	0.0 ± 0.0	0.0 ± 0.0	0.0 ± 0.0	0.0 ± 0.0
Peo 3-sop-5-glu	0.1 ± 0.1 b ^z^	0.1 ± 0.0 b	0.3 ± 0.1 a	0.1 ± 0.0 b	0.2 ± 0.1 ab
Cya 3-p-hydroxybenzoyl sop-5-glu	0.0 ± 0.0 NS	0.0 ± 0.0	0.0 ± 0.0	0.0 ± 0.0	0.0 ± 0.0
Unknown	0.0 ± 0.0 NS	0.0 ± 0.0	0.0 ± 0.0	0.0 ± 0.0	0.0 ± 0.0
Peo 3-p-hydroxybenzoyl sop-5-glu	0.0 ± 0.0 b	0.0 ± 0.0 b	0.1 ± 0.0 b	0.1 ± 0.0 b	1.3 ± 0.2 a
Unknown	0.0 ± 0.0 NS	0.0 ± 0.0	0.0 ± 0.0	0.0 ± 0.0	0.0 ± 0.0
Cya 3-feruloyl-sop-5-glu	0.0 ± 0.0 NS	0.0 ± 0.0	0.0 ± 0.0	0.0 ± 0.0	0.0 ± 0.0
Peo 3-feruloyl-sop-5-glu	0.0 ± 0.0 c	0.0 ± 0.0 c	0.3 ± 0.0 a	0.1 ± 0.0 b	0.0 ± 0.0 c
YGM-2	0.1 ± 0.1 b	0.0 ± 0.0 b	0.3 ± 0.0 a	0.0 ± 0.0 b	0.0 ± 0.0 b
YGM-1b	0.0 ± 0.0 NS	0.0 ± 0.0	0.0 ± 0.0	0.0 ± 0.0	0.0 ± 0.0
YGM-1a	0.0 ± 0.0 NS	0.0 ± 0.0	0.0 ± 0.0	0.0 ± 0.0	0.0 ± 0.0
YGM-5b	0.6 ± 0.2 ab	0.6 ± 0.0 ab	1.7 ± 0.1 a	0.5 ± 0.1 ab	0.0 ± 0.0 b
YGM-3	0.1 ± 0.1 b	0.1 ± 0.0 b	0.3 ± 0.0 a	0.1 ± 0.0 b	0.0 ± 0.0 c
YGM-4b	0.1 ± 0.0 bc	0.1 ± 0.0 cd	0.2 ± 0.0 a	0.1 ± 0.0 b	0.0 ± 0.0 d
YGM-5a	0.6 ± 0.1 cd	0.7 ± 0.0 bc	1.3 ± 0.1 a	1.0 ± 0.1 b	0.4 ± 0.1 d
YGM-6	0.7 ± 0.1 bc	0.5 ± 0.0 c	1.7 ± 0.2 a	0.9 ± 0.1 b	0.9 ± 0.1 d
Total anthocyanin	2.3 ± 0.7 b	2.1 ± 0.1 b	6.2 ± 0.7 a	3.0 ± 0.3 b	1.9 ± 0.4 b

^z^ Each datum represents mean ± S.D. (*n* = 3). Different letters in each row represent significant differences at 5% level as determined by Tukey’s multiple range test. ^y^ DAT: Days after transplant; nurseries of four cultivars were transplanted to Miyazaki research field on 29 May 2021. ^x^ NS: Not significant. Photographs show cross section of roots at 15 DAT, 30 DAT, 45 DAT, 60 DAT, and harvest time. Black bars = 1 mm; white bars = 10 mm.

**Table 2 plants-12-01907-t002:** Effect of growth stages on anthocyanin content and composition in root system of ‘Akemurasaki’.

DAT ^y^	15	30	45	60	Harvest Time
	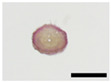	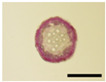	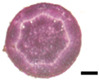	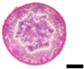	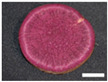
	Anthocyanin content (mg·100 g^−1^ FW)
Cya 3-sop-5-glu	0.1 ± 0.0 c ^z^	0.0 ± 0.0 c	6.7 ± 0.3 b	1.1 ± 0.1 c	29.5 ± 2.3 a
Peo 3-sop-5-glu	0.0 ± 0.0 c	0.2 ± 0.0 c	5.2 ± 0.2 b	1.1 ± 0.1 c	50.2 ± 2.3 a
Cya 3-p-hydroxybenzoyl sop-5-glu	0.1 ± 0.1 d	0.2 ± 0.0 d	11.3 ± 0.3 b	2.0 ± 0.2 c	11.9 ± 0.2 a
Unknown	0.1 ± 0.1 c	0.0 ± 0.0 c	0.6 ± 0.0 b	0.0 ± 0.0 c	2.2 ± 0.1 a
Peo 3-p-hydroxybenzoyl sop-5-glu	0.0 ± 0.0 d	0.3 ± 0.0 d	11.4 ± 0.2 b	2.5 ± 0.2 c	21.1 ± 0.6 a
Unknown	0.0 ± 0.1 bc	0.0 ± 0.0 c	0.2 ± 0.0 b	0.0 ± 0.0 c	3.1 ± 0.1 a
Cya 3-feruloyl-sop-5-glu	0.0 ± 0.0 c	0.0 ± 0.0 c	2.3 ± 0.1 b	0.4 ± 0.0 c	5.8 ± 0.3 a
Peo 3-feruloyl-sop-5-glu	0.0 ± 0.0 c	0.0 ± 0.0 c	1.2 ± 0.1 b	0.2 ± 0.0 c	6.4 ± 0.2 a
YGM-2	1.0 ± 0.1 c	0.7 ± 0.0 c	20.3 ± 3.7 b	3.7 ± 03 c	71.7 ± 3.9 a
YGM-1b	0.2 ± 0.0 c	0.2 ± 0.0 c	3.9 ± 0.2 b	0.6 ± 0.1 c	7.4 ± 0.4 a
YGM-1a	2.1 ± 0.2 d	3.3 ± 0.1 d	46.5 ± 1.4 a	8.1 ± 0.7 c	30.8 ± 1.3 b
YGM-5b	0.5 ± 0.0 c	0.4 ± 0.0 c	12.0 ± 0.6 b	2.5 ± 0.2 c	110.1 ± 5.9 a
YGM-3	0.9 ± 0.1 c	0.7 ± 0.0 c	11.5 ± 0.2 b	1.8 ± 0.1 c	28.3 ± 1.6 a
YGM-4b	0.1 ± 0.0 c	0.1 ± 0.0 c	4.1 ± 0.1 b	0.8 ± 0.1 c	14.9 ± 1.0 a
YGM-5a	1.8 ± 0.1 d	3.9 ± 0.1 d	54.1 ± 0.8 b	10.8 ± 0.9 c	73.2 ± 3.6 a
YGM-6	0.6 ± 0.1 c	0.5 ± 0.0 c	9.3 ± 0.2 b	1.5 ± 0.1 c	34.6 ± 1.8 a
Total anthocyanin	7.6 ± 0.8 c	10.6 ± 0.5 c	200.7 ± 5.4 b	37.0 ± 3.2 c	501.2 ± 23.8 a

^z^ Each datum represents mean ± S.D. (*n* = 3). Different letters in each row represent significant differences at 5% level as determined by Tukey’s multiple range test. ^y^ DAT: Days after transplant; nurseries of four cultivars were transplanted to Miyazaki research field on 29 May 2021. Photographs show cross section of roots at 15 DAT, 30 DAT, 45 DAT, 60 DAT, and harvest time. Black bars = 1 mm; white bars = 10 mm.

**Table 3 plants-12-01907-t003:** Effect of growth stages on anthocyanin content and composition in root system of ‘Purple Sweet Lord’.

DAT ^y^	15	30	45	60	Harvest Time
	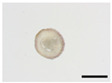	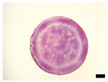	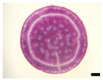	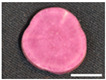	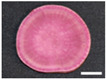
	Anthocyanin content (mg·100 g^−1^ FW)				
Cya 3-sop-5-glu	0.0 ± 0.0 c ^z^	0.3 ± 0.0 c	2.2 ± 0.1 b	3.0 ± 0.5 a	0.3 ± 0.1 c
Peo 3-sop-5-glu	0.0 ± 0.0 d	0.8 ± 0.0 cd	1.7 ± 0.1 c	5.3 ± 0.7 a	3.5 ± 0.3 b
Cya 3-p-hydroxybenzoyl sop-5-glu	0.0 ± 0.0 d	0.4 ± 0.0 cd	2.8 ± 0.1 b	3.6 ± 0.4 a	0.6 ± 0.0 c
Unknown	0.0 ± 0.0 c	0.0 ± 0.0 c	0.4 ± 0.0 b	0.7 ± 0.1 a	0.0 ± 0.0 c
Peo 3-p-hydroxybenzoyl sop-5-glu	0.0 ± 0.0 e	1.3 ± 0.0 d	2.8 ± 0.2 c	5.4 ± 0.5 a	4.1 ± 0.1 b
Unknown	0.0 ± 0.0 c	0.0 ± 0.0 c	0.3 ± 0.0 bc	1.6 ± 0.2 a	0.3 ± 0.1 b
Cya 3-feruloyl-sop-5-glu	0.0 ± 0.0 d	0.1 ± 0.0 d	1.0 ± 0.0 b	2.2 ± 0.3 a	0.4 ± 0.1 b
Peo 3-feruloyl-sop-5-glu	0.0 ± 0.0 c	0.1 ± 0.0 c	0.5 ± 0.1 c	3.0 ± 0.4 a	1.6 ± 0.1 b
YGM-2	0.0 ± 0.0 d	2.4 ± 0.2 c	3.6 ± 0.2 b	6.3 ± 0.8 a	2.0 ± 0.3 c
YGM-1b	0.0 ± 0.0 c	0.8 ± 0.1 b	0.7 ± 0.1 b	1.7 ± 0.2 a	0.1 ± 0.1 c
YGM-1a	0.0 ± 0.0 e	2.6 ± 0.2 c	5.7 ± 0.3 a	4.8 ± 0.6 b	1.2 ± 0.2 d
YGM-5b	0.0 ± 0.0 c	3.2 ± 0.2 b	3.1 ± 0.2 b	12.4 ± 1.4 a	10.5 ± 1.3 a
YGM-3	0.0 ± 0.0 d	1.7 ± 0.1 c	1.8 ± 0.1 c	5.6 ± 0.5 a	2.6 ± 0.4 b
YGM-4b	0.1 ± 0.1 d	2.0 ± 0.1 c	1.1 ± 0.1 c	5.9 ± 0.7 a	3.2 ± 0.5 b
YGM-5a	0.0 ± 0.0 d	5.2 ± 0.3 bc	6.8 ± 0.4 b	10.7 ± 1.2 a	5.0 ± 0.5 c
YGM-6	0.0 ± 0.0 d	2.8 ± 0.1 c	1.7 ± 0.1 cd	12.1 ± 1.1 a	9.6 ± 0.9 b
Total anthocyanin	0.1 ± 0.1 d	23.8 ± 1.4 c	36.0 ± 2.2 bc	84.4 ± 9.2 a	45.1 ± 4.8 b

^z^ Each datum represents mean ± S.D. (*n* = 3). Different letters in each row represent significant differences at 5% level as determined by Tukey’s multiple range test. ^y^ DAT: Days after transplant; nurseries of four cultivars were transplanted to Miyazaki research field on 29 May 2021. Photographs show cross section of roots at 15 DAT, 30 DAT, 45 DAT, 60 DAT, and harvest time. Black bars = 1 mm; white bars = 10 mm.

**Table 4 plants-12-01907-t004:** Effect of growth stages on anthocyanin content and composition in root system of ‘Tanegashimamurasaki’.

DAT ^y^	15	30	45	60	Harvest Time
	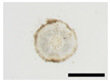	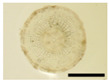	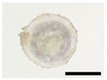	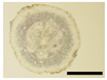	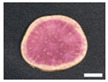
	Anthocyanin content (mg 100 g^−1^ FW)				
Cya 3-sop-5-glu	0.0 ± 0.0 b ^z^	0.0 ± 0.0 b	0.0 ± 0.0 b	0.0 ± 0.0 b	0.6 ± 0.0 a
Peo 3-sop-5-glu	0.0 ± 0.0 NS ^x^	0.0 ± 0.0	0.0 ± 0.0	0.0 ± 0.0	0.0 ± 0.0
Cya 3-p-hydroxybenzoyl sop-5-glu	0.0 ± 0.0 NS	0.0 ± 0.0	0.0 ± 0.0	0.0 ± 0.0	0.0 ± 0.0
Unknown	0.0 ± 0.0 NS	0.0 ± 0.0	0.0 ± 0.0	0.0 ± 0.0	0.0 ± 0.0
Peo 3-p-hydroxybenzoyl sop-5-glu	0.0 ± 0.0 NS	0.0 ± 0.0	0.0 ± 0.0	0.0 ± 0.0	0.0 ± 0.0
Unknown	0.0 ± 0.0 NS	0.0 ± 0.0	0.0 ± 0.0	0.0 ± 0.0	0.0 ± 0.0
Cya 3-feruloyl-sop-5-glu	0.0 ± 0.0 b	0.0 ± 0.0 b	0.0 ± 0.0 b	0.0 ± 0.0 b	0.9 ± 0.1 a
Peo 3-feruloyl-sop-5-glu	0.0 ± 0.0 NS	0.0 ± 0.0	0.0 ± 0.0	0.0 ± 0.0	0.0 ± 0.0
YGM-2	0.1 ± 0.1 b	0.0 ± 0.0 b	0.1 ± 0.0 b	0.2 ± 0.0 b	8.7 ± 0.4 a
YGM-1b	0.0 ± 0.0 b	0.0 ± 0.0 b	0.0 ± 0.0 b	0.0 ± 0.0 b	2.6 ± 0.1 a
YGM-1a	0.0 ± 0.0 b	0.0 ± 0.0 b	0.0 ± 0.0 b	0.0 ± 0.0 b	1.0 ± 0.0 a
YGM-5b	0.0 ± 0.0 b	0.0 ± 0.0 b	0.0 ± 0.0 b	0.0 ± 0.0 b	0.3 ± 0.1 a
YGM-3	0.0 ± 0.0 b	0.0 ± 0.0 b	0.0 ± 0.0 b	0.0 ± 0.0 b	8.6 ± 0.4 a
YGM-4b	0.0 ± 0.0 NS	0.0 ± 0.0	0.0 ± 0.0	0.0 ± 0.0	0.0 ± 0.0
YGM-5a	0.0 ± 0.0 NS	0.0 ± 0.0	0.0 ± 0.0	0.0 ± 0.0	0.0 ± 0.0
YGM-6	0.0 ± 0.0 b	0.0 ± 0.0 b	0.0 ± 0.0 b	0.0 ± 0.0 b	0.7 ± 0.0 a
Total anthocyanin	0.1 ± 0.1 b	0.0 ± 0.0 b	0.1 ± 0.0 b	0.3 ± 0.0 b	23.3 ± 1.1 a

^z^ Each datum represents mean ± S.D. (*n* = 3). Different letters in each row represent significant differences at 5% level as determined by Tukey’s multiple range test. ^y^ DAT: Days after transplant; nurseries of four cultivars were transplanted to Miyazaki research field on 29 May 2021. ^x^ NS: Not significant. Photographs show cross section of roots at 15 DAT, 30 DAT, 45 DAT, 60 DAT, and harvest time. Black bars = 1 mm; white bars = 10 mm.

## Data Availability

Not applicable.

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
