# Peer review of "Effect of Growth Stages on Anthocyanins and Polyphenols in the Root System of Sweet Potato"

_plants, 2023, doi:10.3390/plants12091907_

Round 1
Reviewer 1 Report
The manuscript entitled “Effect of Growth Stages on Anthocyanins and Polyphenols of the Root System Before the Beginning of Storage Root Enlargement of Sweetpotato (Ipomoea batatas (L.) Lam) '', studies the effects of changes in polyphenol and anthocyanin content and composition in the root systems of four sweetpotato cultivars from the early growth stages and examined why acylated anthocyanins and polyphenols need to accumulate in the root systems of purple-fleshed sweetpotato from the early growth stage. They should address the subject and critically review the information from the literature.
Suggestions:
The authors need to revise the title of the paper in a more meaningful way. the title is long and has some unnecessary information;
Please improve the abstract to cover the important topics reviewed and discussed in this article. The abstract is written in a way lacks logic. It should highlight the salient findings more critically;
Keywords are present in the title (Sweetpotato; root system), choose other indexing terms for the article;
The introduction is well written and contextualized. However, as paragraphs are too long, please better;
The results have long paragraphs. I suggest reducing the size of the paragraphs. The results of this study are not fully explained therefore the interpretation of the results is very difficult. The author needs to provide the % increase or decrease rather than just writing ''significantly increased….'';
What do the bars at the observable points in Figure 3 and 4 mean? Standard deviation or standard error?
Authors should discuss the results integrally. The discussion is based on individual results. I suggest that integrating the results will give more value to the work. I suggest that you discuss by integrating all your results. You can use correlation tests (PCA or Pearson Correlation).
The discussion is poorly written hence, needs rewriting. The discussion is it's short and poor. The discussion should be further strengthened by adding some more relevant papers. The literature search is INSUFFICIENT, only few related research papers in the past five years are cited (48%, approximately), add the latest research results appropriately.
The report on M&M is very succinct! Provide experimental work plan at the start of M&M. No detail description is available about the experimental design.
What statistical method is used? Describe in detail!
The conclusion is totally confusing. Re-write the conclusion! It needs to be much improved.
Author Response
For all reviewers:
We are sorry for those awkward grammar and clumsy words in previous manuscript and thank you for reading on to the end of my manuscript patiently. According to your precious comments, I will answer all questions one by one.
Reviewer 1
The manuscript entitled “Effect of Growth Stages on Anthocyanins and Polyphenols of the Root System Before the Beginning of Storage Root Enlargement of Sweetpotato (Ipomoea batatas (L.) Lam) '', studies the effects of changes in polyphenol and anthocyanin content and composition in the root systems of four sweetpotato cultivars from the early growth stages and examined why acylated anthocyanins and polyphenols need to accumulate in the root systems of purple-fleshed sweetpotato from the early growth stage. They should address the subject and critically review the information from the literature.
Row2-4 : The authors need to revise the title of the paper in a more meaningful way. the title is long and has some unnecessary information;
Answer: Thank you for the suggestion. We changed the title to “Effect of Early Growth Stages on Anthocyanins and Polyphenols in the Root System of Sweetpotato”.
Row19-35 : Please improve the abstract to cover the important topics reviewed and discussed in this article. The abstract is written in a way lacks logic. It should highlight the salient findings more critically;
Answer: Thank you for your advice. In this study, we newly discovered the timing of accumulation of acylated anthocyanins in purple-fleshed sweet potato cultivars and the rapid increase in total polyphenol content prior to hypertrophy. As indicated, the abstract was revised as follows.
These data indicate that early growth stage of root system is a critical time. This timing may be a strategy to maximize the accumulation of polyphenols with high antioxidant activity, as well as acylated anthocyanins, to protect against abiotic and biotic stresses.
Row36-37 : Keywords are present in the title (Sweetpotato; root system), choose other indexing terms for the article;
Answer: Thank you for your advice. we chose other indexing terms ‘pigment, histological observation’
Row66: The introduction is well written and contextualized. However, as paragraphs are too long, please better;
Answer: Thank you for your advice. We broke lines and shortened a paragraph.
Row197-201 : The results have long paragraphs. I suggest reducing the size of the paragraphs. The results of this study are not fully explained therefore the interpretation of the results is very difficult. The author needs to provide the % increase or decrease rather than just writing ''significantly increased….'';
Answer: Thank you for your advice. I shortened paragraphs and provided the %
increase or decrease.
For the four cultivars, the values of 45 DAT were approximately 1.4-4.9 times higher than the values of 15 DAT. In particular, a high polyphenol content of 307.3 mg⋅ 100 g− 1 FW was recognized in the root system of 'Akemurasaki'. Subsequently, the total polyphenol content of all cultivars decreased with root growth at harvest time. The decrease rate ranged from 28 to 85%.
Figure 3 and 4 : What do the bars at the observable points in Figure 3 and 4 mean? Standard deviation or standard error?
Answer: Thank you for pointing that out. Bars is standard deviation. I added ‘Each data represent mean ± S.D. (n=3).’
Discussion : Authors should discuss the results integrally. The discussion is based on individual results. I suggest that integrating the results will give more value to the work. I suggest that you discuss by integrating all your results. You can use correlation tests (PCA or Pearson Correlation).
Answer: Thank you for your advice. In this study, three varieties of distinctive purple-fleshed sweet potatoes were selected, including a control 'Kokei No. 14'. Although similar trends were observed for the increase and decrease in total polyphenol content, no significant values were found for the correlations among the other surveyed items (In particular, anthocyanin content and polyphenol content in each variety). The characteristics of individual varieties are considered important in this paper.
Row280-294 : The discussion is poorly written hence, needs rewriting. The discussion is it's short and poor. The discussion should be further strengthened by adding some more relevant papers. The literature search is INSUFFICIENT, only few related research papers in the past five years are cited (48%, approximately), add the latest research results appropriately.
Answer: Thank you for your advice. The discussion text, especially polyphenol biosynthesis (lignin synthesis), was revised and new references were added.
Row361-363, 395-396 : The report on M&M is very succinct! Provide experimental work plan at the start of M&M. No detail description is available about the experimental design.
Answer: Thank you for pointing that out. I provided experimental work plan.
Row 411-413 : What statistical method is used? Describe in detail!
Answer: Thank you for pointing that out. I added '4.5. Statistical analysis' to Materials and Methods.
Row 415-434 : The conclusion is totally confusing. Re-write the conclusion! It needs to be much improved.
Answer: Thank you for the suggestion. I rewrote the conclusion.
Reviewer 2 Report
This article entitled "Effect of Growth Stages on Anthocyanins and Polyphenols of the Root System Before the Beginning of Storage Root Enlargement of Sweetpotato (Ipomoea batatas (L.) Lam)" investigated the timing of anthocyanin accumulation in the root system among several cultivars, and showed that the highest percentage of acylated anthocyanins in three purple-fleshed cultivars among all growth stages was at 15 DAT. Although the content seemed trivial and lacks scientific significance, it would be worth publishing considering the content was in the focus of the topic of special issue.
Author Response
For all reviewers:
We are sorry for those awkward grammar and clumsy words in previous manuscript and thank you for reading on to the end of my manuscript patiently. According to your precious comments, I will answer all questions one by one.
Reviewer 2
This article entitled "Effect of Growth Stages on Anthocyanins and Polyphenols of the Root System Before the Beginning of Storage Root Enlargement of Sweetpotato (Ipomoea batatas (L.) Lam)" investigated the timing of anthocyanin accumulation in the root system among several cultivars, and showed that the highest percentage of acylated anthocyanins in three purple-fleshed cultivars among all growth stages was at 15 DAT. Although the content seemed trivial and lacks scientific significance, it would be worth publishing considering the content was in the focus of the topic of special issue.
Answer: Thank you for the suggestion. It may indeed be trivial, but there are few reports on early growth stages and I believe they need to be presented.
Reviewer 3 Report
It is known that potatoes are a polyploid plant. It would be interesting to investigate the level of ploidy of differentiated tissues during development as an indicator of plant viability.
Expand the histological description, as this aspect is of interest to readers.
It would be nice to define the ROS as an indicator of stress resistance of plants.
The list of references should be brought in line with the requirements of the journal.
Author Response
For all reviewers:
We are sorry for those awkward grammar and clumsy words in previous manuscript and thank you for reading on to the end of my manuscript patiently. According to your precious comments, I will answer all questions one by one.
Reviewer3
It is known that potatoes are a polyploid plant. It would be interesting to investigate the level of ploidy of differentiated tissues during development as an indicator of plant viability.
Row 90-92 : Expand the histological description, as this aspect is of interest to readers.
Answer: Thank you for your advice. I added ‘There is primary cambium, protoxylem, meristem cells around the vessel, and periderm. Sweetpotatoe is a xylem hypertrophic growth pattern.’.
Row 513-524 : It would be nice to define the ROS as an indicator of stress resistance of plants.
Answer: Thank you for your advice. I added more new paper about ROS.
- Apel, K.; Hirt, H. Reactive oxygen species: metabolism, oxidative stress, and signaling transduction. Annu Rev Plant Biol. 2004, 55, 373-399. https://doi.org/10.1146/annurev.arplant.55.031903.141701
- Laloi, C.; Apel, K.; Danon, A. Reactive oxygen signalling: the latest news. Curr. Opin. Plant Biol. 2004, 7, 323-328. https://doi.org/10.1016/j.pbi.2004.03.005
- Gong, Y.; Liu, J.Q.; Xu, M.J.; Zhang, C.M.; Gao, J.; Li, C.G.; Xing, K.; Qin, S. Antifungal volatile organic compounds from Streptomyces setonii WY228 control black spot disease of sweet potato. Appl. Environ. Microbiol. 2022, 88, e02317-21.
- Marone, D.; Mastrangelo, A. M.; Borrelli, G. M.; Mores, A.; Laidò, G.; Russo, M. A; Ficco, D. B. M. Specialized metabolites: Physiological and biochemical role in stress resistance, strategies to improve their accumulation, and new applications in crop breeding and management. Plant Physiol. Biochem. 2022, 172, 48-55. https://doi.org/10.1016/j.plaphy.2021.12.037
- Bauduin, S.; Latini, M.; Belleggia, I.; Migliore, M.; Biancucci, M.; Mattioli, R.; Francioso, A.; Mosca, L.; Funck, D.; Trovato, M. Interplay between proline metabolism and ROS in the fine tuning of root-meristem size in arabidopsis. Plants, 2022, 11, 1512. https://doi.org/10.3390/plants11111512
Referencecs : The list of references should be brought in line with the requirements of the journal.
Answer: Thank you for the suggestion. I brought references in line with the requirements of the journal.